# Rehybridization dynamics into the pericyclic minimum of an electrocyclic reaction imaged in real-time

Y. Liu [1,2,9], D. M. Sanchez [1,3,8,9], M. R. Ware[1], E. G. Champenois[1], J. Yang[1,4,5], J. P. F. Nunes[6,7], A. Attar[4], M. Centurion [6], J. P. Cryan [1], R. Forbes [4], K. Hegazy[1], M. C. Hoffmann [4], F. Ji[4], M.-F. Lin [4], D. Luo [4], S. K. Saha [6], X. Shen [4], X. J. Wang [4], T. J. Martínez [1,3] ✉ & T. J. A. Wolf [1] ✉

Electrocyclic reactions are characterized by the concerted formation and cleavage of both σ and π bonds through a cyclic structure. This structure is known as a pericyclic transition state for thermal reactions and a pericyclic minimum in the excited state for photochemical reactions. However, the structure of the pericyclic geometry has yet to be observed experimentally. We use a combination of ultrafast electron diffraction and excited state wave-packet simulations to image structural dynamics through the pericyclic minimum of a photochemical electrocyclic ring-opening reaction in the molecule α-terpinene. The structural motion into the pericyclic minimum is dominated by rehybridization of two carbon atoms, which is required for the transformation from two to three conjugated π bonds. The σ bond dissociation largely happens after internal conversion from the pericyclic minimum to the electronic ground state. These findings may be transferrable to electrocyclic reactions in general.

Electrocyclic reactions, a subgroup of the class of pericyclic reactions, are important synthetic tools in organic chemistry due to their stereospecificity, i.e., their ability to control the reaction outcome through the stereochemistry of the reactant. Moreover, they are relevant to biological processes such as the biosynthesis of vitamin D in human skin[1]. Their stereospecificity originates from the concerted rearrangement of π and σ electrons through a single, cyclic critical geometry simultaneously making and breaking multiple bonds[2]. These simultaneous rearrangements require the involved π and σ molecular orbitals to exhibit a similar orientation to enable their interaction, which requires the rehybridization of some of the involved atoms.

However, the detailed structure of the critical geometries of electrocyclic and pericyclic reactions has never been directly characterized experimentally due to the lack of methods with suitable sensitivity.

The concerted bond rearrangement can be illustrated by electrocyclic ring-opening in 1,3-cyclohexadiene (CHD)-like molecules. The two electrons participating in the σ bond, which is broken during the reaction (($C_3$–$C_4$), see Fig. 1 for the CHD-derivative α-terpinene), and the 4 π-electrons of a system of two conjugated double bonds (($C_1$=$C_6$) and ($C_9$=$C_5$)) undergo a concerted transformation into a set of three conjugated double bonds. The positions of these double bonds change (($C_3$=$C_1$), ($C_6$=$C_9$), and ($C_5$=$C_4$)) during this process, resulting in

[1]Stanford PULSE Institute, SLAC National Accelerator Laboratory, 2575 Sand Hill Road, Menlo Park, CA 94025, USA. [2]Department of Physics and Astronomy, Stony Brook University, Stony Brook, NY 11790, USA. [3]Department of Chemistry, Stanford University, 333 Campus Drive, Stanford, CA 94305, USA. [4]SLAC National Accelerator Laboratory, 2575 Sand Hill Road, Menlo Park, CA 94025, USA. [5]Center of Basic Molecular Science, Department of Chemistry, Mong Man Wai Building of Science and Technology, S-1027 Tsinghua University, Beijing, China. [6]Department of Physics and Astronomy, University of Nebraska-Lincoln, Theodore Jorgensen Hall 208, 855 N 16th Street, Lincoln, NE 68588, USA. [7]Diamond Light Source, Harwell Science Campus, Fermi Ave, Didcot OX11 0DE, UK. [8]Present address: Design Physics Division, Lawrence Livermore National Laboratory, Livermore, CA, USA. [9]These authors contributed equally: Y. Liu, D.M. Sanchez. ✉e-mail: toddjmartinez@gmail.com; thomas.wolf@slac.stanford.edu

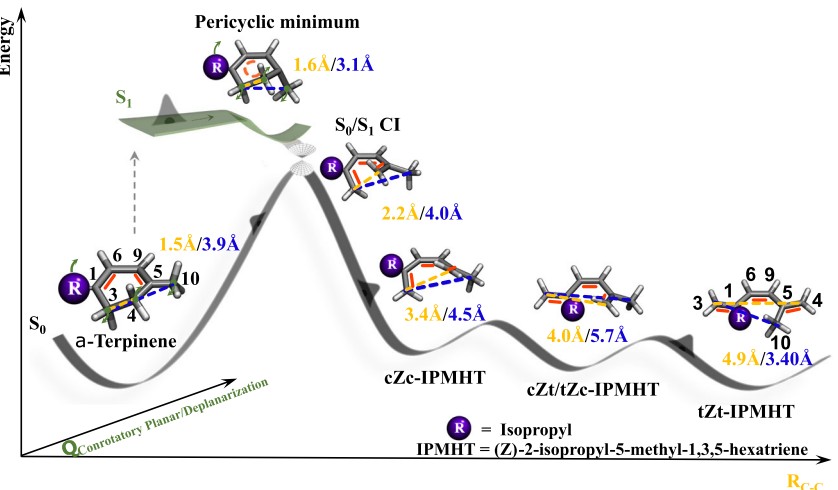

**Fig. 1 | Schematic description of the observed electrocyclic ring-opening dynamics of α-terpinene.** After photoexcitation to the first excited state ($S_1$), the molecule relaxes along a coordinate representing deplanarization with respect to the reactant double bond positions and planarization with respect to the product double bond positions into the pericyclic minimum. The pericyclic minimum is close to, but separated by a shallow barrier from a conical intersection ($S_0/S_1$ CI) with the electronic ground state ($S_0$). Population which relaxes through the CI either returns to the $S_0$ reactant minimum or evolves along a carbon–carbon bond dissociation coordinate $R_{C-C}$ into three $S_0$ minima representing different triene photoproduct isomers labeled with cZc, cZt, tZc, and tZt. Visualizations of representative structures along the reaction coordinate are shown together with specific carbon–carbon distances in yellow and blue. Additionally, the distances are reported by color-coded numbers. Both the structures and the distances are extracted from the simulations. The carbon numbering used in the text is shown in black. The double bond positions are highlighted in the structure visualizations as red bars.

a change in the hybridization of the ($C_3$) and ($C_4$) carbon atoms from $sp^3$ (tetrahedral coordination) to $sp^2$ (planar coordination).

Electrocyclic reactions can proceed through thermal or photochemical pathways with opposite stereospecificity. The famous Woodward–Hoffmann rules predict this behavior based on the symmetry of the involved molecular orbitals and the structural motion from the reactant through the critical geometry to the product[2]. For thermal electrocyclic reactions in the electronic ground state, the critical geometry is referred to as the pericyclic transition state. Due to their short lifetimes, transition states in general can rarely be experimentally characterized, with very few special exceptions[3–6]. Photochemical electrocyclic reactions take place on ultrafast timescales and are enabled by nonadiabatic dynamics through a conical intersection (CI) connecting the lowest excited state ($S_1$) to the ground state ($S_0$, see Fig. 1)[7,8]. Here, the critical geometry represents a minimum in $S_1$ close to the CI, known as the pericyclic minimum[9–11].

The Woodward–Hoffmann rules state that the stereospecificity of the photochemical reaction pathway (which we confirmed in a previous study[12]) is guaranteed by the planarization during the rehybridization of the ($C_3$) and ($C_4$) $CH_2$ groups taking place in a conrotatory fashion, i.e., by rotation of the $CH_2$ groups in the same clockwise or counterclockwise direction. Thus, the pericyclic minimum geometry of CHD represents a critical snapshot of all the above-mentioned simultaneous structural rearrangements: σ bond breaking, rehybridization, π bond alternation, and conrotatory motion.

In a recent, seminal study, the electronic structure changes that take place during the relaxation of $S_1$-excited CHD from the reactant equilibrium geometry to the pericyclic minimum were visualized using time-resolved X-ray absorption spectroscopy[13]. Our present results provide complementary information about the nuclear structure changes during relaxation into the pericyclic minimum using a combination of megaelectronvolt ultrafast electron diffraction and ab initio multiple spawning simulations[12,14–16]. The high structural sensitivity of our methodology provides unprecedented access to the structural details of the critical geometry on the pathway of a photochemical electrocyclic reaction and provides a new perspective on the origins of the reaction's stereospecificity.

We study the α-terpinene (αTP) molecule, which differs from CHD by the addition of methyl and isopropyl substituents (see Figs. 1, 2). The molecule is excited at a wavelength of 267 nm, near the absorption maximum of the $S_1$ state. The photochemical ring-opening of αTP has previously been studied with time-resolved spectroscopy. However, these experiments were only indirectly sensitive to the structural dynamics of the molecule[7,17–21]. According to our experimental and simulation results, the presence of the substituents in αTP does not qualitatively alter the photochemical dynamics in comparison to CHD apart from a slight overall slowing of the dynamics[15]. Hydrogen atoms are difficult to track with electron diffraction because of their weak scattering. The addition of the methyl and isopropyl substituents introduces carbon atoms which act as reporter atoms by adding the stronger signatures of carbon–carbon bond distance changes to our experimental observable, time-dependent atomic pair distribution functions (PDFs). These signatures were missing in previous studies of the structural dynamics of CHD[15,22]. The reporter signatures provide direct evidence for a substantial part of the rehybridization and the conrotatory motion to take place in the $S_1$ state of αTP prior to internal conversion to $S_0$, which triggers σ-bond dissociation.

## Results and discussion

In Fig. 2a, we show the experimental PDF obtained by real-space transformation of static diffraction patterns from gas phase αTP. For comparison, we plot a simulated PDF of the ground state structure of αTP. The simulation is based on a diffraction signal obtained from the initial conditions of our excited state wavepacket simulations (see the "Methods" section for further details). Experimental and simulated PDFs are in quantitative agreement. The main contributors to the PDFs are carbon–carbon distances, although they also contain weak signatures from C–H and H–H distances, which we will neglect in the following discussion (see Supplementary Note 1). The structural information contained in the PDFs is conveniently presented in the framework of carbon–carbon coordination spheres. The carbon–carbon distance distributions extracted from the simulations are additionally shown color-coded with respect to the coordination spheres in Fig. 2a. Representative distances from the first three

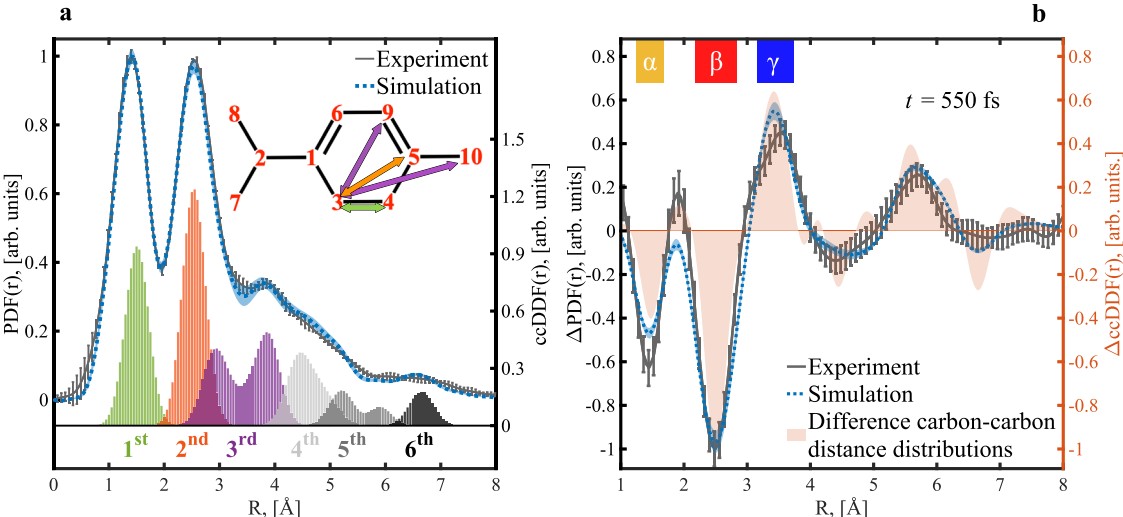

**Fig. 2 | Experimental and simulated structural information of αTP.** The line plots in panel **a** show both the simulated and experimental pair distribution functions (PDFs) of the molecule in the ground state. The histograms below the PDFs represent carbon–carbon distance distribution functions (ccDDF) based on the initial geometries of our ab initio multiple spawning simulations separated and color-coded with respect to carbon coordination spheres. The inset of panel **a** shows the labeling of the carbon atoms of αTP as used in the text. Additionally, representative distances for the first three coordination spheres are marked by color-coded arrows. Panel **b** shows experimental and simulated difference PDF (ΔPDF) at a pump-probe delay of 550 fs. The light-orange-colored area-plot indicates the total difference carbon–carbon distance distribution function (ΔccDDF) from all the carbon coordination spheres. Three regions are labeled as α, β, and γ. Uncertainties (s.e.m.) derived from bootstrapping analyses are shown as error bars (experiment) and shaded areas (simulation).

coordination spheres are shown as color-coded arrows in the inset of Fig. 2a.

The peak at 1.4 Å in the PDFs can be associated with the first coordination sphere (green) representing carbon–carbon bond distances. The largest contributions to the peak at 2.5 Å originate from the second coordination sphere (orange), distances between carbon atoms bonded to the same carbon (e.g., $(C_3,C_5)$). The 2.5 Å maximum exhibits a shoulder towards larger distances due to contributions from the third carbon coordination sphere (purple, distances between carbon atoms connected to a common carbon–carbon bond, e.g. $(C_3,C_{10})$). As noted previously[12], the third carbon coordination sphere has a bimodal distance distribution in rigid ring systems due to cis (e.g. $(C_3,C_9)$) and trans (e.g. $(C_3,C_{10})$) conformations about the central bond. The broad tail of the PDFs towards distances beyond 4 Å results from distances in higher coordination spheres.

A difference PDF (ΔPDF), which is the difference between delay-dependent PDF and static PDF, obtained 550 fs after photoexcitation, when the ring-opening is already completed, is shown in Fig. 2b. Due to their differential nature and as demonstrated by the difference carbon–carbon distance distributions (orange in Fig. 2b), a distance change appears as a combination of a negative contribution to the ΔPDF at the original distance and a positive contribution at the distance, which is reached at the delay time of the ΔPDF. The ΔPDF shows the strongest features in distance regimes labeled with α, β, and γ in Fig. 2b. The α and β regions closely resemble the positions of the first and second coordination sphere in Fig. 2a. Since the α and β signals exhibit negative amplitudes, they correspond to the disappearance of distances in the reactant first and second coordination sphere. Such signatures are only consistent with ring-opening[12,15]. The bond dissociation as part of the ring opening increases the $(C_3,C_4)$ distance and, therefore, leads to a negative signature in the α region. The β region is dominated by negative signatures from increases in the $(C_1,C_4)$ and $(C_3,C_5)$ distances. All three of these distances increase during the ring-opening reaction towards values in the γ regime and beyond, leading to a positive signature there. The ΔPDF exhibits additional negative and positive signatures in the 4–5 and 5–6 Å regions which result from ring-opening induced (C,C) distance changes in the third and higher coordination spheres.

We simulate the ring-opening dynamics of αTP using ab initio multiple spawning (AIMS)[23–25] in combination with α-state-averaged complete active-space self-consistent field theory (α-CASSCF)[26] for electronic structure determination and generate ΔPDFs from them (see Fig. 2b, Figs. 3–5 and the see the "Methods" section). We did not observe any dependence of the dynamics in the AIMS simulations on the three rotamers of αTP and, therefore, use in the following the average of the simulated signatures from all individual rotamers (see Supplementary Note 2). The observed excited state lifetime is in good agreement with previous spectroscopic studies[7,20]. According to our simulations, 58% of the αTP excited state population relaxing through the conical intersection with $S_0$ undergoes electrocyclic ring-opening, whereas the remaining population returns to the reactant minimum. We find a high level of agreement between experimental and simulated ΔPDFs. Figure 3a shows in detail the temporal onset of the ΔPDF signal from Fig. 2b over several delay steps around time zero. The α, β, and γ regions are highlighted in Fig. 3a. The time-dependent evolution of the integrated signal from the three regions is plotted in Fig. 3b. We fit the temporal onset of the transient signal in all three regions with error functions (see Supplementary Note 3). The center and width of these error function fit are shown in Fig. 3b. Both experiment and simulation show a delayed onset of the α and β signatures with respect to the γ signature around time zero (see onset times and arrows in Fig. 3b).

We have assigned the positive amplitude in the γ region of the temporal snapshot of Fig. 2b at 550 fs delay to an increase of the $(C_3,C_4)$, $(C_1,C_4)$, and $(C_3,C_5)$ and other distances from the α and β to the γ regime. This assignment cannot hold for the early onset of the positive signature in γ at time zero since it precedes the onset of the corresponding negative α and β signatures. Thus, the signature must originate from structural dynamics prior to the $(C_3–C_4)$ bond breaking and the structural opening of the ring.

We have observed in previous studies of similar rigid ring systems a collapse of the bimodal distribution of the third coordination sphere. This is due to the redistribution of the absorbed photon energy during non-adiabatic dynamics lowering the molecular rigidity and leading to significant out-of-plane motions[12,27]. The latter increases the third coordination sphere distances of carbons in *cis*-configuration (see above) and decreases the third coordination sphere distances of

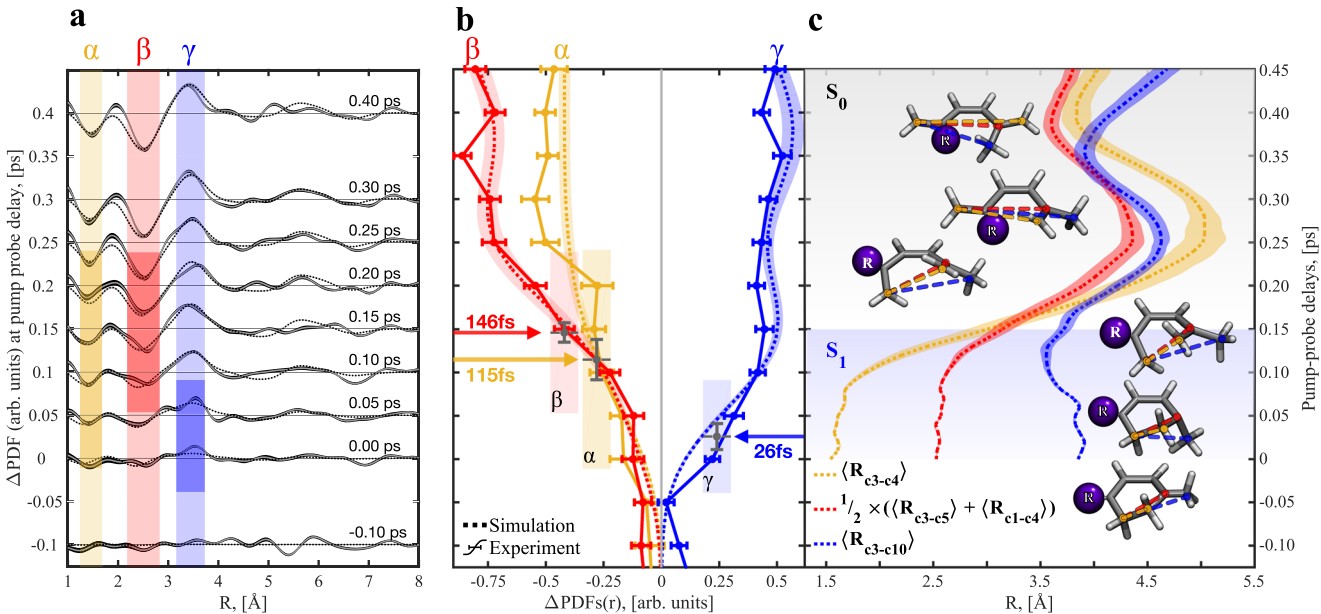

**Fig. 3 | Comparison of measured and simulated time-dependent difference pair distribution functions (ΔPDFs).** Panel **a** shows experimental and simulated (solid and dotted black lines) ΔPDFs at different delays in a time window of 0.5 picoseconds (ps) around time-zero. Simulations use the ab initio multiple spawning (AIMS) methods (see the "Methods" section). Analogous to Fig. 2, α, β, and γ regions are highlighted with different colors. Panel **b** shows the integrated experimental (solid) and simulated (dotted) time-dependent intensity of the three regions from panel **a** with identical color-coding. We fit error functions to the temporal onsets of the experimental signals (see Supplementary Note 3). The delay values of the error function centers are marked with arrows and as black dots with error bars. Additionally, the widths of the signal onsets according to the error function fits are

marked by color-coded bars in panels (**a**) and (**b**). The shaded areas (simulation) and error bars (measurement) of the line plots indicate the uncertainty (s.e.m.) obtained from bootstrapping analysis (68% confidence interval). For the simulations, these error bars reflect convergence with respect to initial condition sampling. The temporal evolution of three representative carbon–carbon distances in the AIMS simulations, the $(C_3-C_4)$, $(C_3,C_5)$, and $(C_3,C_{10})$ distances, (labeling according to Fig. 2a) is plotted in panel (**c**). Additionally, snapshots of the molecular geometry evolution based on a representative AIMS trajectory are shown with the three representative carbon–carbon distances marked. Note the alignment of the vertical axes in all three panels.

carbons in *trans*-configuration. The corresponding signatures in a ΔPDF are negative peaks at the positions of the two *cis* and *trans* maxima (~3 and ~4 Å, respectively) of the third coordination sphere (see Fig. 2a) and a positive peak in the gap between the maxima of the third coordination sphere (3.4 Å), overlapping with the observed γ signature. However, the early signature in the γ region, as observed in the present study, agrees only partially with this expectation: We observe a clear positive signature at 3.4 Å and a weak negative signature at 4 Å, which is close to the noise level in the experimental data, but clearly visible in the simulations (see Supplementary Fig. 1). However, a corresponding negative signature at smaller distances, around 3 Å, is missing. Thus, the early onset of the γ signature must exclusively originate from a distance reduction of larger third coordination sphere distances in *trans*-configuration (see above) and distances from higher coordination spheres. An exclusive reduction of third coordination sphere distances in *trans*-configuration can only be consistent with out-of-plane motion of the $(C_{10})$ reporter carbon of the methyl substituent (e.g., $(C_3,C_{10})$). The observed effect, a distance decrease from 4 to 3.4 Å, cannot be caused by a shrinking of the $(C_5-C_{10})$ bond distance. Such a motion would significantly shorten the bond distance far into the repulsive part of its potential. Additionally, it would generate a corresponding signature with similarly early onset in the α-region of the ΔPDF. An in-plane shrinkage of the $(C_3,C_{10})$ distance over several bonds would again also cause decreases of third coordination sphere distances in *cis*-configurations. Out-of-plane motion of the isopropyl group would also reduce distances in *cis*-configuration (e.g., $(C_8,C_6)$) which is not supported by our data.

Our simulations give further evidence for such a motion as visualized at the example of the $(C_3,C_{10})$ distance in Fig. 3c. At 100 fs after photoexcitation, the methyl ring substituent shows significant out-of-plane displacement (see Fig. 4c, green for the out-of-plane angle

and structures in Fig. 3c), which leads to a reduction of the $(C_3,C_{10})$ distance from 3.9 to 3.55 Å while the $(C_3-C_4)$ and the $(C_3,C_5)$ distances do not yet show substantial displacements. The $(C_3-C_4)$ distance only shows considerable enlargement after 150 fs. With the $(C_3-C_4)$ distance increase, the $(C_3,C_{10})$ pair contributes to the ΔPDF at higher distances. The contributions to the γ regime are taken over by the $(C_3-C_4)$, $(C_3,C_5)$, and other carbon pairs not highlighted in Fig. 3c. All three mean distances plotted in Fig. 3c experience a minimum in the 350–400 fs range. However, this effect is washed out in both the experimental and simulated ΔPDFs (see Supplementary Fig. 1) due to the width of the distribution and the presence of additional carbon–carbon distances in the same region. Thus, we observe in both experiment and simulation a clear temporal separation between the methyl group out-of-plane bending, which leads to the early rise of the γ signature in the ΔPDFs, and the structural opening of the ring, which leads to the delayed onset of the α and β signatures in the ΔPDFs.

Figure 4a shows a projection of the excited state (blue) and ground state (black) components of the trajectory representations of the simulated wavepacket onto the $(C_3-C_4)$ distance. The projection clearly shows that $(C_3-C_4)$ bond dissociation happens exclusively in the ground state and quasi-instantaneously after population transfer to the ground state through the CI (see Fig. 1). Thus, the methyl group out-of-plane bending must take place in the excited state prior to internal conversion through the conical intersection with the ground state. Hence, it is a direct signature of the structural relaxation to the pericyclic minimum of $S_1$.

The out-of-plane bending can be directly related to conrotatory rehybridization dynamics enabling interaction between π and σ electrons of the molecule. Rehybridization of the $(C_3)$ and $(C_4)$ $CH_2$ groups from $sp^3$ to $sp^2$ hybridization must lead to a planarization around the terminal double bonds of the triene photoproduct $((C_3=C_1)$ and

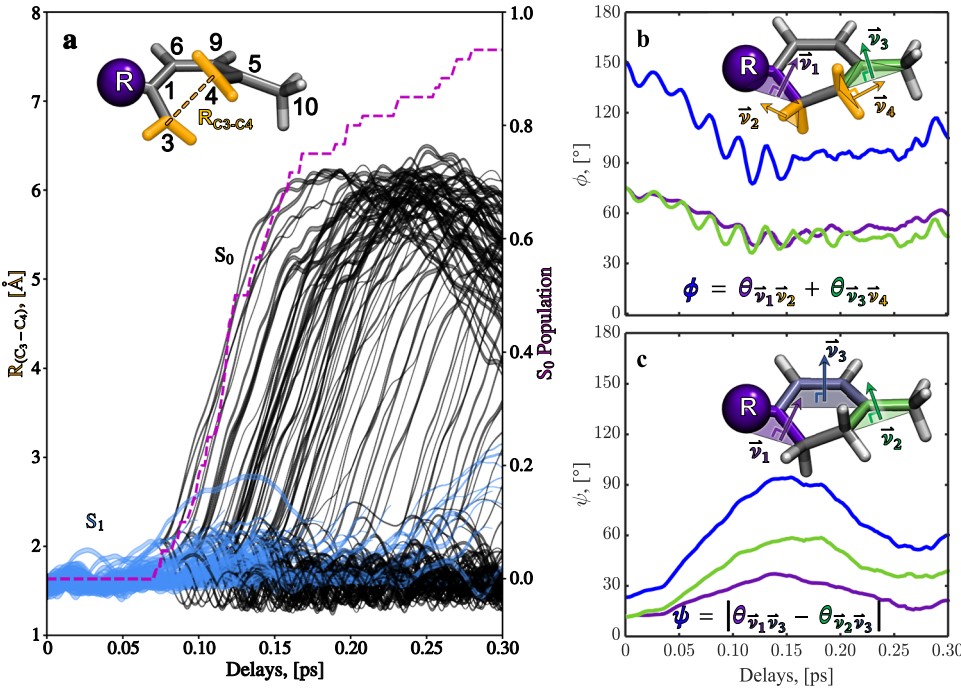

**Fig. 4 | Temporal evolution of the simulated trajectories in several nuclear degrees of freedom.** Panel **a** shows the evolution of the ($C_3$–$C_4$) distance in the excited state ($S_1$, blue) and the ground state ($S_0$, black). Additionally, the time-dependent population of $S_0$ is plotted (pink). Bond dissociation, i.e. ring-opening, happens directly after internal conversion from $S_1$ to $S_0$. The atom labeling is shown in the inset. Panel **b** shows the time-dependent expectation value of the projection of the simulated nuclear wavepacket evolution in $S_1$ onto the conrotatory planarization coordinate $\phi$ (blue). The coordinate is defined in the inset and represents the conrotatory addition of the angles between the plane defined by the ($C_3$) $CH_2$

group ($\vec{v_2}$) and the plane defined by the $C_1$, $C_2$, and $C_3$ carbons ($\vec{v_1}$, purple plot), and between the planes defined by the ($C_4$) $CH_2$ group ($\vec{v_4}$) and the plane defined by the $C_4$, $C_5$, and $C_{10}$ carbons ($\vec{v_3}$, green curve), respectively. The corresponding projections onto a conrotatory deplanarization coordinate $\psi$ (blue) are plotted in panel (**c**). The coordinate is defined in the inset and represents the conrotatory addition of the angles of the planes defined by the $C_1$, $C_2$, and $C_3$ carbons ($\vec{v_1}$, purple curve) and the $C_4$, $C_5$, and $C_{10}$ carbons ($\vec{v_2}$, green curve) with respect to a common plane defined by the $C_1$, $C_6$, $C_9$, and $C_5$ ($\vec{v_3}$).

($C_4$=$C_5$), see Fig. 1), i.e. moving the ($C_3$) $CH_2$ group into a common plane with the ($C_1$), ($C_2$), and ($C_6$) carbons and the ($C_4$) $CH_2$ group into a common plane with the ($C_5$), ($C_9$), and ($C_{10}$) carbons, respectively. Such a planarization could be achieved by the conrotatory movement of the ($C_3$) and ($C_4$) hydrogens around the respective carbons, in line with the simplified picture given by Woodward and Hoffmann[2]. However, it is strongly restricted by the presence of the still intact ($C_3$–$C_4$) $\sigma$ bond in $S_1$ (see Supplementary Note 4).

As an alternative, planarization with respect to the terminal double bonds of the triene photoproduct can be achieved by deplanarization of the methyl and isopropyl substituents with respect to the conjugated ($C_1$=$C_6$) and ($C_9$=$C_5$) double bonds of the reactant. In Fig. 4b, c, we plot the expectation value of the excited state component of the simulated wavepacket onto the corresponding degrees of freedom. Figure 4b depicts a projection onto the conrotatory planarization coordinate $\phi$ with respect to the terminal ($C_1$=$C_3$) and ($C_4$=$C_5$) double bonds of the triene photoproduct involving the methyl and isopropyl substituents. The projection of the excited state wavepacket onto the complementary conrotatory deplanarization coordinate $\psi$ with respect to the *cis*-butadiene-like conjugated double bond system of the reactant is plotted in Fig. 4c. The simulated excited state wavepacket shows substantial evolution in both the planarization and deplanarization coordinates and confirms the out-of-plane motion to be dominated by the methyl group (see additional details in Supplementary Note 5). Additionally, the minimum (maximum) points of the motions in Fig. 4b (4c) temporally coincide well with depopulation to the electronic ground state (see the pink curve in Fig. 4a). This finding strongly suggests a connection between the deplanarization/planarization motion, i.e., $\pi$ bond alternation and $CH_2$

rehybridization, and the access to the conical intersection seam in the vicinity of the pericyclic minimum.

The upper row of Fig. 5 shows two-dimensional projections of the simulated excited state wavepacket density (red contour lines) onto the ($C_3$–$C_4$) distance and the deplanarization angle $\psi$ from Fig. 4c at different delay times (see Supplementary Movie 1 for an animated version of Fig. 5). Corresponding projections onto the ($C_3$–$C_{10}$) distance and $\psi$ are plotted in the lower row of Fig. 5. Specifically at the onset of population transfer to the ground state (100 fs delay), the projection shows significant deformation from an initially round shape into the diagonal direction due to an anti-correlation between the ($C_3$–$C_{10}$) distance and $\psi$ (lower row), i.e., a ($C_3$–$C_{10}$) distance decrease correlated with a $\psi$ increase. In contrast, there is neither strong correlation (distance increase with angle increase) nor anti-correlation for the ($C_3$–$C_4$) distance and $\psi$ (upper row). Additionally, the significant motion of the excited state wavepacket density from outside into the $\gamma$ regime of the ΔPDFs (blue-shaded area, as defined in Fig. 3) can be seen in the lower-row graphs, whereas the density maximum of the excited wavepacket barely leaves the $\alpha$-regime (yellow-shaded area, upper row). Thus, the early onset of the amplitude increase within the $\gamma$ regime of the experimental ΔPDFs (Fig. 3) can be regarded as a unique and sensitive gauge for the conrotatory deplanarization in the molecule and, thus, for the rehybridization of the ($C_3$) and ($C_4$) $CH_2$ groups.

We compare the wavepacket evolution in Fig. 5 with two significant points identified in our theoretical investigations of the $S_1$ potential energy surface, the pericyclic minimum geometry (red cross in Fig. 5 and molecular geometry in Fig. 1) and the minimum energy conical intersection geometry (MECI, green circle in Fig. 5 and molecular geometry in Fig. 1). The MECI is separated from the pericyclic minimum by a small barrier. Both geometries show significant out-of-

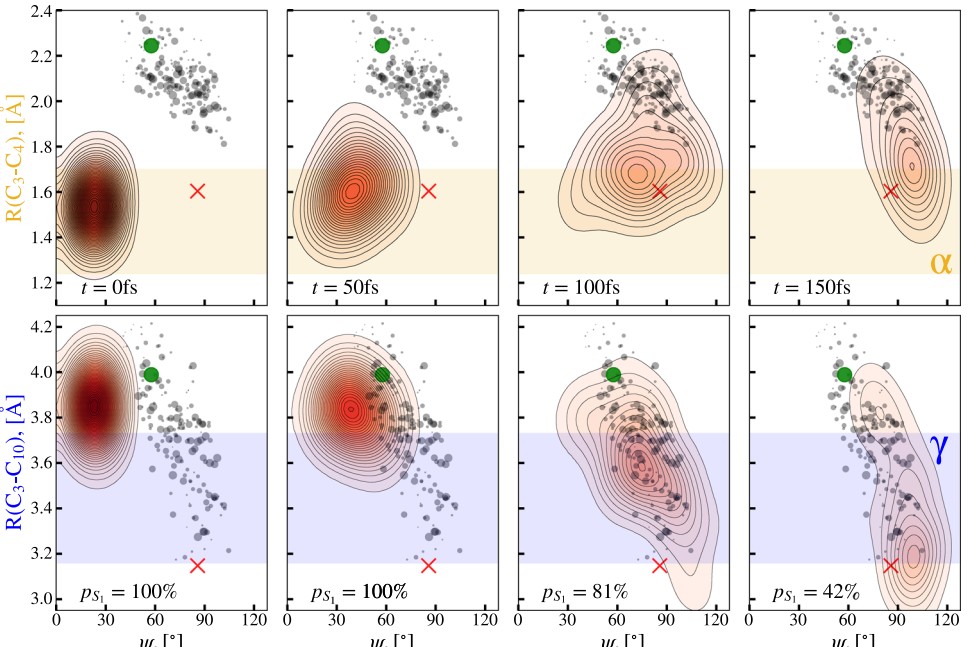

**Fig. 5 | Two-Dimensional projections of the simulated excited state wavepacket density.** Projections (red contours) onto the $(C_3–C_4)$ distance and the conrotatory deplanarization angle $\psi$ from Fig. 4c for different time delays are shown in the upper row. Analogous projections onto the $(C_3,C_{10})$ distance and $\psi$ are depicted in the lower row. The delays for each column are written in the upper-row plots, the population fraction residing in the excited state is marked in the lower-row plots.

For comparison, the $\alpha$ and $\gamma$ areas of Fig. 3 are marked in yellow and blue. Additionally, the geometries at which population transfer to the ground state ultimately takes place are shown as gray circles with sizes proportional to the relative amount of transferred population. The minimum energy conical intersection geometry is marked as a green circle and the geometry of the pericyclic minimum as a red cross. For an animated version of Fig. 5 see Supplementary Movie 1.

plane bending. The wavepacket motion in both projections of Fig. 5 is clearly driven by a gradient in the Franck–Condon region of $S_1$ (0 fs delay in Fig. 5) towards the pericyclic minimum. In the vicinity of the pericyclic minimum, it encounters a region of the $S_1$ potential energy surface with strong nonadiabatic coupling leading it to undergo internal conversion to $S_0$ (gray circles in Fig. 5). Thus, Fig. 5 nicely demonstrates that the conical intersection seam is the origin of the strong non-adiabatic coupling which drives internal conversion and subsequent ring-opening, but that nonadiabatic transitions do not necessarily happen exactly at the conical intersection seam or the MECI.

In conclusion, by the combination of ultrafast electron diffraction and AIMS simulations, we provide a detailed molecular picture of the rehybridization dynamics to the critical geometry on the photochemical pathway of an electrocyclic reaction, the pericyclic minimum. The pericyclic minimum represents different levels of progress for the multiple concerted processes involved in the reaction, $\sigma$ bond dissociation, $\pi$-bond alternation, and rehybridization. We observe a significant level of rehybridization and $\pi$-bond alternation happening during relaxation in $S_1$ toward the pericyclic minimum. However, the structural motion during the relaxation can only be explained by rehybridization in the presence of an intact $\sigma$-bond. Thus, the pericyclic minimum represents an early stage of the reaction with respect to $\sigma$-bond dissociation and a significantly later stage with respect to the other processes. Our results provide a new perspective on the origins of the stereospecificity of electrocyclic reactions: The stereoconfiguration of the triene photoproduct is preserved by excited state rehybridization dynamics in the presence of the $\sigma$ bond locking the double bond structure of the triene photoproduct in place rather than by a conrotatory motion of the $CH_2$ groups during the $\sigma$ bond dissociation. The observed relative timing between rehybridization and $\sigma$-bond dissociation is in principle disfavored by the specific structure of $\alpha$TP as compared

to, e.g., CHD since it requires the out-of-plane motion of a methyl group instead of a much lighter hydrogen atom. Thus, its observation in $\alpha$TP points to our findings constituting an intrinsic property of electrocyclic reactions in general and not specifically of $\alpha$TP. The presence of the methyl reporter group of $\alpha$TP in our chosen experimental observable, ultrafast electron diffraction, merely enables its investigation.

## Methods

### Experimental setup

The experimental apparatus is described in detail elsewhere[28,29]. In short, we use the 800 nm, 50 fs output of a Ti:Sapphire laser system and separate two beam paths. Pulses in both beam paths are frequently tripled. The pulses of the probe beam path are directed onto the photocathode of an RF gun and eject an ultrashort pulse containing ~$10^4$ electrons. 4.2 MeV electrons are generated using an S-band photocathode radio frequency (RF) gun[29] and focused to a spot size of 200 μm FWHM in the interaction region of a gas phase experimental chamber. The pump pulses (4–6 μJ) are focused into the experimental chamber to a diameter of 240 μm FWHM and overlapped with the electron pulses at a 1° angle through a holey mirror in the interaction region of a gas phase experimental chamber. The experimental response function including effects of the optical and electron pulse length as well as relative arrival time jitter is estimated to be 150 fs[14,15]. $\alpha$-Terpinene ($\alpha$TP, purity >95%) is purchased from Sigma-Aldrich and used without further purification. We employ a static-filled 3 mm flow cell (550 μm orifices, sample at room temperature) in combination with a repetition rate of 360 Hz. Diffracted electrons are detected by a combination of a phosphor screen and an EMCCD camera. Based on the relative static and dynamic signal levels, we estimate that we excite about 1.56% of the molecules (for details see Supplementary Note 6). Time-dependent diffraction is measured as a series of time delay points between −2 and +2 ps in each scan. The separation between time delay points is 50 fs, except for the earliest and latest delay points,

where it is considerably larger. At each time delay point, we integrate the diffraction signal for 10 s (3600 shots), and the full data set includes 145 such scans at each delay. The sequence of delay steps is randomized for every scan to avoid systematic errors. The processing of the obtained raw diffraction patterns is described in detail in Supplementary Note 6.

## Excited-state dynamics simulations

Ab initio multiple spawning (AIMS) simulations interfaced with GPU-accelerated α-complete active space self-consistent field theory (α-CASSCF)[30–32] are used to model the photochemical ring-opening dynamics of αTP in isolation. In its ring-closed form, αTP exhibits three relatively stable and optically indistinguishable rotamers that differ by the orientation of the isopropyl group with respect to the ring plane. For this reason, all are included (with equal weight) in the AIMS simulations using identical methods. For more details about the different structural signatures of the rotamers see Supplementary Note 2. Our active space consists of six electrons in four orbitals ($1\sigma^*$, $2\pi$, and $1\pi^*$) and the two lowest energy singlet states (referred to as $S_0$ and $S_1$ for the ground and 1st excited state in the adiabatic representation, respectively), within the 6-31G* basis set using an $\alpha$ value of 0.82, i.e. $\alpha(0.82)$-SA2-CAS(6,4)SCF/6-31G*. Based on our previous work with 1,3-cyclohexadiene (CHD) and α-phellandrene (αPH), it is expected that this level of theory is well-suited for describing the photoinduced ring-opening dynamics of αTP[12,15]. Similar to CHD and αPH, there exist both closed- and ring-open $S_0/S_1$ conical intersections (CI) in αTP, which correspond to geometries where the CHD moiety puckers out of plane (OOP) or the $C_3$–$C_4$ bond (Fig. 2a and Supplementary Fig. 2) elongates to ~2.2 Å, respectively. Geometries, energies, and CI vectors of the minimum energy conical intersections and other stationary points on the potential energy surface of αTP are included in Supplementary Data 1. Supplementary Fig. 2 shows the critical points (i.e., stationary points) upon evolving away from the Frank–Condon point on $S_1$ along the open and closed photochemical pathways for the three rotamers of αTP. As in CHD and αPH, the ring-open CI is lower in energy than the closed-ring CI, offering some insight into a branching ratio that heavily favors the ring-opening pathway (see discussion in Supplementary Note 7). In addition, we observe a minimum on $S_1$ near the ring-open CI, where the methyl group moves the OOP of the CHD moiety and the $C_3$–$C_4$ bond elongates to 1.6 Å. This is consistent with the pericyclic minimum in photoinduced electrocyclic reactions: an intermediate on $S_1$ that leads to ring-opening dynamics via non-radiative decay through the ring-open CI. All α-CASSCF electronic structure calculations are performed with the TeraChem electronic structure package[33–35].

Supplementary Fig. 3a shows an ultraviolet (UV) electronic absorption spectrum generated from 300 geometries (100 per αTP rotamer) sampled from a harmonic Wigner distribution corresponding to the rotamer's PBE0/6-31G* ground state optimized structure. Single point energy calculations are performed at the $\alpha(0.82)$-SA2-CAS(6,4)SCF/6-31G* level of theory and the resulting $S_0 \rightarrow S_1$ excitation energies homogeneously broadened using an oscillator-strength scaled Gaussian with full-width half maximum (FWHM) of 0.2 eV. The active-space molecular orbitals (MO) at the $S_0$ minima are shown in Supplementary Fig. 3b for all rotamers, which are nearly identical to those used in our CHD and αPH work. A total of 60 initial conditions (20 sets of positions and momenta for each rotamer) are selected to initiate the AIMS dynamics. These initial conditions (ICs) are selected under the constraint that their $S_0 \rightarrow S_1$ transition energy was within 0.3 eV of the pump pulse (4.65 eV) used in the experiment after applying a 0.4 eV red-shift to the theoretical spectrum to align the theoretical and experimental absorption

maxima. These initial conditions are then placed on the $S_1$ surface and propagated with AIMS.

AIMS expands the full molecular wavefunction into a time-dependent basis of multi-dimensional frozen-width Gaussian functions that evolve along adiabatic PESs according to the time-dependent molecular Schrödinger equation.

$$\chi_I(R,t) = \sum_{k=1}^{N_I(t)} c_k^I(t)\chi_k^I(R;\underline{R}_k^I(t),\underline{P}_k^I(t),\underline{\gamma}_k^I(t),\alpha_k^I) \tag{1}$$

where $N_I(t)$ represents the total number of trajectory basis functions (TBFs) on electronic state $I$, $c_k^I(t)$ is the time-dependent complex coefficient of the $k$th TBF, $\alpha_k^I$ is the frozen TBF width, and $\chi_k^I(\dots)$ is a multidimensional frozen Gaussian that is expressed as a product of one-dimensional Gaussian functions corresponding to the 3N nuclear degrees of freedom. The reader is referred elsewhere for a more detailed description of performing and analyzing excited-state dynamics simulations within the AIMS framework[12,15,23–25]. The first two singlet states ($S_0$ and $S_1$) are included in the AIMS dynamics. All required electronic structure quantities (energies, gradients, and nonadiabatic couplings) are calculated on-the-fly with α-SA2-CAS(6,4)SCF/6-31G*. An adaptive timestep of 0.48 fs (20 au) (reduced to 0.12 fs (5 au) in regions with large nonadiabatic coupling) is used to propagate the centers of the trajectory basis functions (TBFs). A coupling threshold of 0.01 au (scalar product of nonadiabatic coupling and velocity vectors) demarcates spawning events generating new TBFs on different electronic states. Population transfer between TBFs is described by solving the time-dependent Schrödinger equation in the time-evolving TBF basis set.

We simulate the ultrafast dynamics for the first 1 ps of all three rotamers of αTP by: (1) using AIMS to propagate the initial wavepacket for the first 500 fs or until all population has returned to the ground state, (2) stopping TBFs on the ground state when they are decoupled from other TBFs (off-diagonal elements of the Hamiltonian become small), and (3) adiabatically continuing these stopped TBFs using the positions and momenta from the last frame in AIMS as initial conditions for adiabatic molecular dynamics with unrestricted DFT using the Perdew–Burke–Ernzerof hybrid exchange-correlation functional[36], i.e., uPBE0/6-31G*. A total of 234 TBFs are propagated, with 174 of these being adiabatically continued on the ground state with DFT. The simulation of time-dependent $\Delta PDF(r, t)$ is described in detail in Supplementary Note 6.

## Data availability

The raw diffraction image data generated in this study have been deposited in the Zenodo repository under the accession code https://doi.org/10.5281/zenodo.7570707[37]. The data from the ab initio multiple spawning simulations have been deposited in the Zenodo repository under accession code https://doi.org/10.5281/zenodo.7854203[38].

## Code availability

The data analysis codes are available from the corresponding authors upon request.

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

## Acknowledgements

We thank Markus Gühr, Dennis Mayer, and Stephen Weathersby for their support of the experiment and helpful discussions. This work was supported by the AMOS program within the U.S. Department of Energy, Office of Science, Basic Energy Sciences, Chemical Sciences, Geosciences, and Biosciences Division. MeV-UED is operated as part of the Linac Coherent Light Source at the SLAC National Accelerator Laboratory, supported in part by the U.S. Department of Energy (DOE) Office of Science, Office of Basic Energy Sciences, SUF Division Accelerator and Detector R&D program, the LCLS Facility, and SLAC under contract Nos. DE-AC02-05CH11231 and DE-AC02-76SF00515. Lawrence Livermore National Laboratory is operated by Lawrence Livermore National Security, LLC, for the U.S. Department of Energy, National Nuclear Security Administration, under Contract DE-AC52-07NA27344. J.P.F.N. and M.C. were supported by the US Department of Energy Office of Science, Basic Energy Sciences under award no. DE-SC0014170.

## Author contributions

Y.L., M.R.W., E.G.C., J.Y., J.P.F.N., A.A., M.C., J.P.C., R.F., K.H., M.C.H., F.J., M.-F.L., D.L., S.K.S., X.S., X.J.W., and T.J.A.W. prepared and conducted the experiment at the SLAC ultrafast electron diffraction facility. D.M.S. and T.J.M. performed the ab-initio simulations. Y.L. analyzed the experimental data. Y.L., D.M.S., T.J.M., and T.J.A.W. interpreted the

results. Y.L., D.M.S., and T.J.A.W. wrote the manuscript. All authors discussed the science of the paper.

## Competing interests

The authors declare no competing interests.
