## [Peer Review File · Nature Communications]

REVIEWER COMMENTS

Reviewer #1 (Remarks to the Author):

This manuscript reports the latest progress of mapping the chemical reaction dynamics in real time using a direct structural probe of ultrafast electron diffraction (UED). The reaction is the photochemical electrocyclic reaction in the molecule α -terpinene, which involves concerted formation and cleavage of both σ and π bonds through a cyclic structure. Previous studies revealed that the reaction begins by motion into the pericyclic minimum and the electronic structure changes during this motion were visualized using time-resolved X-ray absorption spectroscopy. However, the detailed structure of the critical geometries of electrocyclic and pericyclic reactions have never been directly characterized experimentally. This work, combining the UED and ab initio multiple spawning simulations, provides a detailed molecular picture of the rehybridization dynamics to the pericyclic minimum for the first time, where a significant level of rehybridization and π -bond alternation happening during relaxation in S_1 towards the pericyclic minimum with molecular level details are observed. This observation provides a new perspective on the origins of the stereospecificity of electrocyclic reactions and has important implication in electrocyclic reactions in general. The manuscript is well written.

Overall, this work is of sufficient interest and validity, suitable for the readership of a high profile journal.

Reviewer #2 (Remarks to the Author):

Rehybridization dynamics...

Liu et al.

This remarkable paper reports a study of ultrafast photochemical dynamics in α -terpinene achieved through a combination of diffraction by a relativistic electron beam and nonadiabatic dynamics by the AIMS method. It is simply astonishing how much is revealed by this powerful combination, and the authors, to my mind, clearly deliver on the promise of the title and more. The

electron diffraction data itself is limited and it is only possible to see the electronic dynamics hidden therein through the calculations. The latter employ a semiempirical variant of CASSCF that appears to be well-validated by the authors for these purposes although I am not in a position to judge that. The key observations from the theory are that changes in the pi system on S1 precede sigma bond fission which takes place after transfer to S0. This is made possible by deplanarization of the methyl substituent rather than simple conrotatory motion of the CH2 groups. The paper certainly merits publication in Nature Comm. I have a few questions and suggestions to improve the clarity.

I suggest they mention the photoexcitation wavelength early on and note it is near the S1 absorption maximum, rather than in passing as they discuss what a difference PDF is. Rather than referring to the "photoproduct" they might explicitly state the transformation leads ultimately to a "triene photoproduct" in which case the necessary nuclear rearrangements can be easily envisaged prior to the more detailed discussion of planarization and deplanarization.

They choose 550 fs to show the ring-opened system on S1 in Fig. 1 but shorter times are generally shown elsewhere, for example Fig. 4. It seems the C3-C4 distance is dropping again after ring opening, also suggested by Fig. 3C. Perhaps they could comment on this.

The conformer heterogeneity was a concern for me until I saw it clearly discussed in the SM. I think they it would be useful to mention this in the main text.

In the Fig 5 caption I suggest they add red with contours... population transfer ultimately takes place...

Again for the casual reader it would help to note planarization (CH2) and deplanarization (methyl) and planarization/deplanarization motions.

Finally, in the abstract they assert that the findings may be transferable to electrocyclic reactions in general but it seems there is no further discussion on that point in the text except perhaps the concluding sentence or two. I think this key point merits some additional elaboration.

Reviewer #3 (Remarks to the Author):

Liu et al report a beautiful investigation on photoinduced electrocyclic reactions, combining ultrafast electron diffraction with nonadiabatic dynamical simulations to image the reaction in real-time. As a case study they choose the photochemical electrocyclic ring opening reaction of a derivative of cyclohexadiene. The presence of the methyl and isopropyl substituents help to enhance the electron diffraction signal. The goal of the authors is to observe in real-time the nuclear motion and in particular the passage through the pericyclic minimum. The main experimental observable is the Carbon-Carbon pair distribution function and its difference in time. The increase and decrease of the distribution height corresponding to the first, second and third coordination spheres are interpreted to trace the change of the molecular structure. The nonadiabatic dynamical simulation remarkably agree with experiment and provide complementary information, helping to provide a complete picture of the reaction.

I think this is a remarkable piece of work and deserves publication in Nature Communications.

However in my opinion there are a number of concerns that should be dealt within before accepting the manuscript.

I found the manuscript a bit difficult to read. I admit this is partly due to the complexity and the originality of the study and so it is a necessary "difficulty". Partly this is also due to the compact format of the journal. However I think the authors may do some additional effort to improve the clarity, thus helping the reader.

1) My major problem is with figures. They are aesthetically beautiful and it is clear they have been deeply meditated and planned, I cannot deny it, but they are probably too ambitious. On one side the information content is really (maybe too) dense, on the other side there are, in some cases, unnecessary embellishments. All these things obstacle a bit in my opinion a rapid understanding by the reader, especially in printed form.

1a) Figure 1: The orange and yellow number are hard to read (mostly on the paper); Understanding the colour code for the two distances reported in orange/blue took me really a while. Since there is no explanation in the caption, a reader should notice the two tiny dashed lines on the structures of the pericyclic minimum.

The long red line on the minimum is probably a symbol for the photoexcitation, but it is really not necessary. It also has the same colour as the bars indicating the double bond (of course nobody would think it is a bond, but it is only adding complexity in my opinion).

The spot reporting the UED signal is not commented in the caption. As far as I can understand it is there mostly for aesthetical reasons, but again it is adding complexity. Moreover, it took me a while to notice that there is a very pale green cone starting from the molecular structure to the signal, and probably the colour is chosen to make apparent that it is an electron diffraction signal since also the “electron” is green. Are all these things really necessary?

1b) Figure 2. Left panel: the colour code of the histograms should be explained by the tiny arrows on the molecular structure. Their colour is difficult to catch, especially for the one superimposed on the C3-C4 bond.

I could not understand why the right y-axis is orange. For sure there is a colour code here, but, as far as I can see, it is not explained.

The caption indicates that in the right panel there are “shaded areas” on the lines reporting the simulation results. I had to consult the PDF version of the manuscript to notice them. They are so thin that I’m wondering if it is really necessary to show them, engaging the reader with the research of such details. Maybe this information could be reported in more readable scale in the Supporting Information

1c) Figure 3. Admittedly this is simpler to read. Still, it was not evident to me from the very beginning that in all three panels the y axis has a meaning since it is reporting time and, of course, with a coherent scale. Once I digested this info, I could appreciate that the figure is beautiful and planned with a lot of care. But, again, the first feeling was a bit arduous.

Moreover, only after analysing the figure few times I noticed that in the rightmost panel the colour code of the three lines correspond to distances traced on the small molecular structure reported on the right-top corner. Again I had to consult the pdf version to notice the orange dashed line on rightmost CC bond.

2) If I properly understood the text, the sentence “Both experimental and simulation...time zero” at page 8 refers to the behaviour of the lines in Figure 3b, close to 0, probably in the 0-50 fs range. Is there a physical information in the fact that the curvature of the experimental and simulated blue lines seem to be different in the -0.05-0.05 range? Is it due to experimental resolution/time duration of the pulses? Why does the simulated line start deviating from zero even before time zero?

3) The discussion in the first half of page 9, is crucial. I had to read it few times before getting it. The point, as far as I understand, is the fact that is missing the negative feature at 3 Angstrom corresponding to the “cis” arrangement. Therefore, this is a feature corresponding only to the “trans” configuration. Now the two carbons that only have “trans” carbons in position 4 are C10 and C2. The authors conclude it must be a movement of C10 and it must be an out of plane movement because, probably, they don’t observe a feature corresponding to a marked elongation of the C5-C10 bond. The possibility of a movement of C2 is discarded because it would lead to a decrease of the “cis” C8-C6 and C3-C7...A very nice but also complicated inference. Maybe the reader would be helped by a less compact discussion.

I’d try to reword the paragraph and make it a bit clearer. Moreover, since we are speaking of curvilinear motions, it does not seem self-evident to me to what extent, such out-of-plane motion would affect the distances of the non-bonded carbons (I mean it is clear if they increase or decrease but not to what extent), and if such “extent” could have been detected by the experiment or not (the argument adopted to discard a motion of C2). Maybe some further comments (also in the SI) and a Table with some data (CC distances) along the trajectory might be helpful to give the reader a feeling on which typical displacement we are discussing.

Moreover, it seems to me that the out of-plane of the C2 is discarded making the hypothesis that it moves as a rigid body. In principle, it is possible that opposite movements of C8 and C7 might compensate the supposed shortening of the distances. Is it true? Can they be ruled out? Are they ruled out on the grounds of the results of the simulation?

Finally, I have to highlight that along all the text the authors very often use atom labels, and the only figure that assisted me to understand this discussion and many other ones, is the small chemical structure with labels reported as an inset of Figure 2.

4) It took me some time to understand also the second part of page 9, starting with “Our simulations..”. The authors mention a decrease of the distance from 3.9 to 3.2 Ang. After a while I understood this should refer to the very small deviation toward the left of the blue line in Figure 3c at times from 0.05 to 0.1 ps. As far as I can see from the figure, that line does not seem to go below 3.5 Ang, and this was confusing me.

More in general, since practically this is the only detail of panel 3c discussed in the text, I’m wondering if the authors could not choose a different scale better helping the readers to follow their discussion.

5) Figure 4 left panel: Once printed the blue lines are hardly distinguished from the black (gray?) lines. It is maybe a problem of my printer, but I recommend the authors to better check the readability of the figure in printed form.

6) Do I understand properly that the authors have no experimental clue on the timescale of the internal conversion? The information that it takes at least 100 fs only comes from simulations, am I right? How confident can we be of this prediction? Some additional discussion is maybe worthy.

7) In Figure 4C, the definition of the planarization and deplanarization coordinates in the insets of Figure 4 b and c is a bit hard to follow. This is partly due to the fact that the atom labels are not reported and should be consulted in Figure 2, the vectors are small and their direction not perfectly appreciable on that small scale.

Moreover, the variable colours of the symbols of the sums defining ϕ and ψ are maybe chosen to suggest something that may be not captured by the reader.

I do understand that behind these figures there is a lot of work and that is hard to do better. However, at least I suggest the readers to exploit also SI, taking more space and providing larger (and maybe different) figures to help the reader to understand these definitions

8) At the beginning of page 13, I suggest the authors to be more explicit on what they mean by “correlation” and “anti-correlation” between the two considered structural parameters.

9) As far as I understand, in Figure 5, the grey spots indicate points at which the nonadiabatic transition takes for different trajectories (Gaussian WP). I don't understand how it is possible that these points are all outside the region visited by the wavepacket (the red contour lines).

10) The conclusion reported in the last three sentences of the manuscript is really interesting. Am I right if I state that, at the end, this conclusion is reached only thanks to the nonadiabatic simulations? I mean simulations are fully consistent with the available experimental observables, but the experimental observables by themselves have not a unique interpretation leading to the reported conclusion. Right?

This could be made a bit more evident, and maybe one-two lines could be added to discuss if this leaves some uncertainty in the conclusions (e.g. if possible alternative interpretation exist).

In summary this is a beautiful work and I strongly support its publication in Nature Comm. The authors made a lot of work to compact the text and increase as much as possible the information carried by the beautiful figures. I think in some cases they were too ambitious, and the figures are a bit hard to digest. Along the same lines, maybe, compatibly with the space, some additional, a bit

more didactic, discussion might make the paper more accessible to a wide readership. These changes can be done in a minor revisions.

A typo: Page 6 “largerdistances”

Fabrizio Santoro

Response to reviewer comments

We would like to thank all reviewers for their reports and their strong support for publication of a properly revised manuscript. Below we address the reviewers' specific comments point by point. Reviewer comments are formatted in *italic*. Our responses are in normal format. Text of the manuscript is marked as **blue**. Changes based on the reviewers' comments are highlighted as **strikedthrough red** if text is removed. Added text is highlighted in **green**. Both the **strikedthrough red** and **green** highlights are also used in the marked-up version of the revised manuscript. The page and paragraph numbers referenced here refer to the marked-up version of the revised manuscript. We made additional changes to the original version of the manuscript to match the nature communications formatting requirements

Reviewer #1 (Remarks to the Author):

*This manuscript reports the latest progress of mapping the chemical reaction dynamics in real time using a direct structural probe of ultrafast electron diffraction (UED). The reaction is the photochemical electrocyclic reaction in the molecule α -terpinene, which involves concerted formation and cleavage of both σ and π bonds through a cyclic structure. Previous studies revealed that the reaction begins by motion into the pericyclic minimum and the electronic structure changes during this motion were visualized using time-resolved X-ray absorption spectroscopy. However, the detailed structure of the critical geometries of electrocyclic and pericyclic reactions have never been directly characterized experimentally. This work, combining the UED and *ab initio* multiple spawning simulations, provides a detailed molecular picture of the rehybridization dynamics to the pericyclic minimum for the first time, where a significant level of rehybridization and π -bond alternation happening during relaxation in S_1 towards the pericyclic minimum with molecular level details are observed. This observation provides a new perspective on the origins of the stereospecificity of electrocyclic reactions and has important implication in electrocyclic reactions in general. The manuscript is well written.*

Overall, this work is of sufficient interest and validity, suitable for the readership of a high profile journal.

We thank referee 1 for their support.

Reviewer #2 (Remarks to the Author):

Rehybridization dynamics...

Liu et al.

This remarkable paper reports a study of ultrafast photochemical dynamics in α -terpinene achieved through a combination of diffraction by a relativistic electron beam and nonadiabatic dynamics by the AIMS method. It is simply astonishing how much is revealed by this powerful combination, and the authors, to my mind, clearly deliver on the promise of the title and more. The electron diffraction data itself is limited and it is only possible to see the electronic dynamics

hidden therein through the calculations. The latter employ a semiempirical variant of CASSCF that appears to be well-validated by the authors for these purposes although I am not in a position to judge that. The key observations from the theory are that changes in the pi system on S1 precede sigma bond fission which takes place after transfer to S0. This is made possible by deplanarization of the methyl substituent rather than simple conrotatory motion of the CH2 groups. The paper certainly merits publication in Nature Comm. I have a few questions and suggestions to improve the clarity.

We thank the referee for their support and their detailed comments on our manuscript.

- *I suggest they mention the photoexcitation wavelength early on and note it is near the S1 absorption maximum, rather than in passing as they discuss what a difference PDF is.*

We thank the referee for their suggestion and add the following sentence to the second paragraph on page 4:

The molecule is excited at a wavelength of 267 nm, near the absorption maximum of the S₁ state.

Additionally, we modify the sentence in the last paragraph of page 5, where we had mentioned the excitation wavelength previously:

A difference PDF (Δ PDF), which is the difference between delay-dependent PDF and static PDF, obtained 550 fs after photoexcitation ~~by a femtosecond pulse with a wavelength of 266 nm~~, when the ring-opening is already completed, is shown in Fig. 2b.

- *Rather than referring to the “photoproduct” they might explicitly state the transformation leads ultimately to a “triene photoproduct” in which case the necessary nuclear rearrangements can be easily envisaged prior to the more detailed discussion of planarization and deplanarization.*

We agree with the referee that the term “triene” can improve the clarity of the text. We add it in the following places:

- Page 9, last paragraph: Rehybridization of the (C₃) and (C₄) CH₂ groups from sp³ to sp² hybridization must lead to a planarization around the terminal double bonds of the triene photoproduct ((C₃=C₁) and (C₄=C₅), see Fig. 1), i.e. moving the (C₃) CH₂ group into a common plane with the (C₁), (C₂), and (C₆) carbons and the (C₄) CH₂ group into a common plane with the (C₅), (C₉), and (C₁₀) carbons, respectively.
- Page 10, first paragraph: As an alternative, planarization with respect to the terminal double bonds of the triene photoproduct can be achieved by deplanarization of the methyl and isopropyl substituents with respect to the conjugated (C₁=C₆) and (C₉=C₅) double bonds of the reactant. In Fig. 4b and c, we plot the expectation value of the excited state component of the simulated wavepacket onto the corresponding degrees of freedom. Figure 4b depicts a projection onto the conrotatory planarization coordinate ϕ with respect to the terminal (C₁=C₃) and (C₄=C₅) double bonds of the triene photoproduct involving the methyl and isopropyl substituents.

- Page 12, first paragraph: The stereoconfiguration of the triene photoproduct is ~~rather~~ preserved by excited state rehybridization dynamics in the presence of the σ bond locking the double bond structure of the triene photoproduct in place rather than by a conrotatory motion of the CH_2 groups during the σ bond dissociation.
 - Caption to Figure 1: Population which relaxes through the CI either returns to the S_0 reactant minimum or evolves along a carbon-carbon bond dissociation coordinate $R_{\text{C-C}}$ into three S_0 minima representing different triene photoproduct isomers.
- *They choose 550 fs to show the ring-opened system on S1 in Fig. 1 but shorter times are generally shown elsewhere, for example Fig. 4. It seems the C3-C4 distance is dropping again after ring opening, also suggested by Fig. 3C. Perhaps they could comment on this.*

We believe that referee 2 is not referring to Fig. 1, but to panel b in Fig. 2, where we show a difference pair distribution function at a delay of 550 fs. We also believe that referee 2 meant S_0 instead of S_1 , since there is no indication from our own experimental and simulated results or from the literature that substantial amounts of population should be in the S_1 state after 550 fs. The time delay of 550 fs is chosen to be well outside the timescales of excited state depopulation and ring-opening dynamics, i.e., to represent the signature of the triene photoproduct after completion of the ring-opening reaction. Thus, the time delay outside the delays shown e.g. in Fig. 4 is not by mistake but chosen on purpose. Figure S1 demonstrates that at delay times beyond the first 400 fs, the difference pair distribution functions do neither show strong time-dependence in the experimental nor in the simulated results. Thus, we could have chosen a snapshot at a slightly different time delay in Fig. 2b or even the average of several time delays after 400 fs without any significant change to the information content of Fig. 2b. We will, therefore, keep the time step of 550 fs.

The referee is correctly pointing out that the C3-C4 distance drops again after the ring-opening. This is an effect which we already observed in a previous study on the ring-opening of cyclohexadiene (ref. 15 of the manuscript) and originates from rotational motion of the ends of the photoproduct carbon chain triggered by the kinetic energy release into the ring-opening nuclear degree of freedom. Other than in ref. 15, we do not observe a clear signature of this motion in our experimental or simulated difference pair distribution functions. The reason is apparent from Fig. 4a: The wavepacket broadens more strongly for terpinene than in the case of cyclohexadiene. The distance reduction of the leading edge of the wavepacket averages out to a large extent with the larger distances in the trailing edge of the wavepacket. The distance reduction is still visible to some extent in Fig. 3c since it represents the expectation value of the distance. However, Fig. 3c represents only a few selected carbon-carbon distances of the molecule. The effect is basically completely obscured in the difference pair distribution functions by the presence of multiple additional carbon-carbon distances with values in the same region. This is the reason for the missing time-dependence in the experimental and simulated difference pair distribution functions in Fig. S1.

To clarify this point in the manuscript, we add the following sentence to a paragraph stating on the bottom of page 8:

All three mean distances plotted in **Fig. 3c** experience a minimum in the 350-400 fs range. However, this effect is washed out in both the experimental and simulated Δ PDFs (see **Supplementary Figure 1**) due to the width of the distribution and the presence of additional carbon-carbon distances in the same region.

- *The conformer heterogeneity was a concern for me until I saw it clearly discussed in the SM. I think they it would be useful to mention this in the main text.*

We agree with referee 2 that a more explicit mention of this fact in the main text is warranted. Therefore, we added the following sentence to the last paragraph on page 6 of the manuscript:

We did not observe any dependence of the dynamics in the AIMS simulations on the three rotamers of α TP and, therefore, use in the following the average of the simulated signatures from all individual rotamers (see **Supplementary Note 2**).

- *In the Fig 5 caption I suggest they add red with contours... population transfer ultimately takes place...*

We agree with referee 2 that this clarification is necessary and modify the caption of Fig. 5 in the following way:

Projections (red contours) onto the (C_3 - C_4) distance and the conrotatory deplanarization angle ψ from **Fig. 4c** for different time delays are shown in the upper row. Analogous projections onto the (C_3 , C_{10}) distance and ψ are depicted in the lower row. The delays for each column are written in the upper row plots, the population fraction residing in the excited state is marked in the lower row plots. For comparison, the α and γ areas of **Fig. 3** are marked in yellow and blue. Additionally, the geometries at which population transfer to the ground state ultimately takes place are shown as grey circles with sizes proportional to the relative amount of transferred population. The minimum energy conical intersection geometry is marked as a green circle and the geometry of the pericyclic minimum as a red cross.

- *Again for the casual reader it would help to note planarization (CH2) and deplanarization (methyl) and planarization/deplanarization motions.*

We apologize, but we do not fully understand the referee's suggestion.

- *Finally, in the abstract they assert that the findings may be transferable to electrocyclic reactions in general but it seems there is no further discussion on that point in the text except perhaps the concluding sentence or two. I think this key point merits some additional elaboration.*

We agree with referee 2 that some additional discussion of this point is warranted. Therefore, we add the following sentences to the end of the "Results and discussion" section:

The observed relative timing between rehybridization and σ -bond dissociation is in principle disfavored by the specific structure of α Tp as compared to, e.g., CHD, since it requires the out-of-plane motion of a methyl group instead of a much lighter hydrogen atom. Thus, its observation in α Tp points to our findings constituting an intrinsic property of electrocyclic reactions in general and not specifically of α Tp. The presence of the methyl “reporter” group of α Tp in our chosen experimental observable, ultrafast electron diffraction, merely enables its investigation.

Reviewer #3 (Remarks to the Author):

Liu et al report a beautiful investigation on photoinduced electrocyclic reactions, combining ultrafast electron diffraction with nonadiabatic dynamical simulations to image the reaction in real-time. As a case study they choose the photochemical electrocyclic ring opening reaction of a derivative of cyclohexadiene. The presence of the methyl and isopropyl substituents help to enhance the electron diffraction signal. The goal of the authors is to observe in real-time the nuclear motion and in particular the passage through the pericyclic minimum. The main experimental observable is the Carbon-Carbon pair distribution function and its difference in time. The increase and decrease of the distribution height corresponding to the first, second and third coordination spheres are interpreted to trace the change of the molecular structure. The nonadiabatic dynamical simulation remarkably agree with experiment and provide complementary information, helping to provide a complete picture of the reaction.

I think this is a remarkable piece of work and deserves publication in Nature Communications.

We thank the referee for their endorsement of our work.

However in my opinion there are a number of concerns that should be dealt within before accepting the manuscript.

- I found the manuscript a bit difficult to read. I admit this is partly due to the complexity and the originality of the study and so it is a necessary “difficulty”. Partly this is also due to the compact format of the journal. However I think the authors may do some additional effort to improve the clarity, thus helping the reader.*

We apologize for the difficulties with the compact text and figures of the manuscript. We believe we have addressed these shortcomings now with the substantial changes to the figures and the text which we outline below.

- My major problem is with figures. They are aesthetically beautiful and it is clear they have been deeply meditated and planned, I cannot deny it, but they are probably too ambitious. On one side the information content is really (maybe too) dense, on the other side there are, in some cases, unnecessary embellishments. All these things obstacle a bit in my opinion a rapid understanding by the reader, especially in printed form.*

- *Figure 1: The orange and yellow number are hard to read (mostly on the paper); Understanding the colour code for the two distances reported in orange/blue took me really a while. Since there is no explanation in the caption, a reader should notice the two tiny dashed lines on the structures of the pericyclic minimum.*

The long red line on the minimum is probably a symbol for the photoexcitation, but it is really not necessary. It also has the same colour as the bars indicating the double bond (of course nobody would think it is a bond, but it is only adding complexity in my opinion).

The spot reporting the UED signal is not commented in the caption. As far as I can understand it is there mostly for aesthetical reasons, but again it is adding complexity. Moreover, it took me a while to notice that there is a very pale green cone starting from the molecular structure to the signal, and probably the colour is chosen to make apparent that it an electron diffraction signal since also the "electron" is green. Are all these things really necessary?

We appreciate the referee's suggestions. We tried to improve Fig. 1 in the following ways (the revised figure is shown below):

- We increased the font size for the orange and yellow numbers and changed the color of the orange numbers to black.

- We modify the caption to specifically point out the lines and numbers:

Figure 1. Schematic description of the observed electrocyclic ring-opening dynamics of α -terpinene. After photoexcitation to the first excited state (S_1), the molecule relaxes along a coordinate representing deplanarization with respect to the reactant double bond positions and planarization with respect to the product double bond positions (red bars) into the pericyclic minimum. The pericyclic minimum is close to, but separated by a shallow barrier from a conical intersection (S_0/S_1 CI) with the electronic ground state (S_0). Population which relaxes through the CI either returns to the S_0 reactant minimum or evolves along a carbon-carbon bond dissociation coordinate R_{C-C} into three S_0 minima representing different triene photoproduct isomers. Visualizations of representative structures along the reaction coordinate are shown together with specific carbon-carbon distances in yellow and blue. Additionally, the distances are reported by color-coded numbers. Both the structures and the distances are extracted from the simulations. The carbon numbering used in the text is shown in black. The double bond positions are highlighted in the structure visualizations as red bars.

- We removed the red line indicating photoexcitation from the left-most structure in Fig. 1.

- We removed the electron diffraction pattern, the light-green cone, and the line symbolizing the electron beam from the figure.

- 1b) Figure 2. Left panel: the colour code of the histograms should be explained by the tiny arrows on the molecular structure. Their colour is difficult to catch, especially for the one superimposed on the C3-C4 bond. I could not understand why the right y-axis is orange. For sure there is a colour code here, but, as far as I can see, it is not explained. The caption indicates that in the right panel there are “shaded areas” on the lines reporting the simulation results. I had to consult the PDF version of the manuscript to notice them. They are so thin that I’m wondering if it is really necessary to show them, engaging the reader with the research of such details. Maybe this information could be reported in more readable scale in the Supporting Information

We agree with the referee on almost all of their remarks. We change the figure in the following ways (the revised figure is shown below):

- We change the color code for the first three coordination spheres in panel a and change the references in the text accordingly (page 5, second paragraph): The peak at 1.4 Å in the PDFs can be associated with the first coordination sphere (cyan/green) representing carbon-carbon bond distances. The largest contributions to the peak at 2.5 Å originate from the second coordination sphere (blue/orange), distances between carbon atoms bonded to the same carbon (e.g. (C₃,C₅)). The 2.5 Å maximum exhibits a shoulder towards larger distances due to contributions from the third carbon coordination sphere (pink/purple, distances between carbon atoms connected to a common carbon-carbon bond, e.g. (C₃,C₁₀)).
- We increase the size of the molecular structure in panel a.
- We increase the thickness of the arrows on top of the molecular structure and change them to solid lines in panel a.

- We offset the arrow from the C3-C4 bond axis in panel a to make both equally visible.
- We remove the color coding from the right y-axis of panel a. The color coding is consistent with the color coding of the right y-axis of panel b. However, we fully agree with the referee that this does not make too much sense.
- We refrain from removing the – admittedly thin – shaded areas marking errors. We believe that this is significant information which must be included in the main figures of the manuscript. The fact that they are difficult to see in some, but not all, areas of the plot is in our opinion significant information by itself.

- 1c) Figure 3. Admittedly this is simpler to read. Still, it was not evident to me from the very beginning that in all three panels the y axis has a meaning since it is reporting time and, of course, with a coherent scale. Once I digested this info, I could appreciate that the figure is beautiful and planned with a lot of care. But, again, the first feeling was a bit arduous.

Moreover, only after analysing the figure few times I noticed that in the rightmost panel the colour code of the three lines correspond to distances traced on the small molecular structure reported on the right-top corner. Again I had to consult the pdf version to notice the orange dashed line on rightmost CC bond.

The referee makes very helpful observations which we are trying to address with the following modifications of the figure (the revised figure is shown below):

- The y-axes are in fact not completely the same in all three panels. In panel a, not only temporal information is reported there, but also the intensity of the plotted difference pair distribution functions. We emphasize this by giving them different labels. We tried to make it more clear that the y axes of panels b and c are the same by joining them together. Additionally, we added the following sentence to the end of the figure caption:

Note the alignment of the vertical axes in all three panels.

- We optimized the horizontal scale of panel c.

- We increased the size of the structures shown in panel c and of the color-coded lines.

- 2) If I properly understood the text, the sentence “Both experimental and simulation...time zero” at page 8 refers to the behaviour of the lines in Figure 3b, close to 0, probably in the 0-50 fs range. Is there a physical information in the fact that the curvature of the experimental and simulated blue lines seem to be different in the -0.05-0.05 range? Is it due to experimental resolution/time duration of the pulses? Why does the simulated line start deviating from zero even before time zero?

We believe that this is a misunderstanding. The delays in onset are much larger than the 50 fs timescale referee 3 talks about and are marked with color-coded numbers and arrows in Fig. 3b. We believe that this misunderstanding can be avoided by explicitly referring to them in the sentence. Therefore, we modify it accordingly (page 7, top paragraph):

Both experiment and simulation show a delayed onset of the α and β signatures with respect to the γ signature around time zero (see onset times and arrows in Fig. 3b).

The deviations between experiment and simulation in the blue curve of Fig. 3b around time zero are difficult to fully understand based on our study. Some of them are arguably outside the statistical error bars, which point to small contributions from systematic disagreements. We speculate that both the experiment and the simulation contribute some level of systematic errors to these disagreements including the slightly negative numbers before time zero. However, we would like to emphasize that these small discrepancies within an overall almost quantitative agreement between experiment and simulation in no way affect the validity of our results and their interpretation.

- 3) The discussion in the first half of page 9, is crucial. I had to read it few times before getting it. The point, as far as I understand, is the fact that is missing the negative feature at 3 Angstrom corresponding to the “cis” arrangement. Therefore, this is a feature corresponding only to the “trans” configuration. Now the two carbons that only

have “trans” carbons in position 4 are C10 and C2. The authors conclude it must be a movement of C10 and it must be an out of plane movement because, probably, they don’t observe a feature corresponding to a marked elongation of the C5-C10 bond. The possibility of a movement of C2 is discarded because it would lead to a decrease of the “cis” C8-C6 and C3-C7...A very nice but also complicated inference. Maybe the reader would be helped by a less compact discussion.

I’d try to reword the paragraph and make it a bit clearer. Moreover, since we are speaking of curvilinear motions, it does not seem self-evident to me to what extent, such out-of-plane motion would affect the distances of the non-bonded carbons (I mean it is clear if they increase or decrease but not to what extent), and if such “extent” could have been detected by the experiment or not (the argument adopted to discard a motion of C2). Maybe some further comments (also in the SI) and a Table with some data (CC distances) along the trajectory might be helpful to give the reader a feeling on which typical displacement we are discussing.

We are delighted that referee 3 understood our argument for the C10 out of plane bending despite the probably too compact explanation for this conclusion. However, we agree that it would benefit from some more details. For clarity, we’d like to correct referee 3’s rewording of our argument in one small aspect. An alternative explanation to out-of-plane bending for the distance reduction in the third coordination shell distances would be a shrinking of the C5-C10 bond, not an elongation. The bond would have to shrink considerably, by several tenths of an Angstrom which would move it into the repulsive region of basically any covalent bond. Therefore, it can be excluded.

Regarding the referee’s remark about how large the effect would need to be to be detected by the experiment: We believe the relevant information the referee seeks are in principle already in the manuscript. However, we agree that it can be explained better. The required size of the distance change required to be causing the experimental signature is clear from the position of the early feature in the difference pair distribution functions (~ 3.4 Angstroms see Fig. 3a) and the position of the trans-maximum in the static pair distribution functions (4 Angstroms, see Fig. 2a). The simulations agree with such a change in the C₃-C₁₀ distance (see Fig. 3c). The out-of-plane bending angle corresponding to such a distance decrease (almost 60 degrees) is shown in Fig. 4c, green.

We hope that we make these additional connections and, thus, address the referee’s comment by the following additions to the text (starting page 7, last paragraph):

We have observed in previous studies of similar rigid ring systems a collapse of the bimodal distribution of the third coordination sphere. This is due to the redistribution of the absorbed photon energy during non-adiabatic dynamics lowering the molecular rigidity and leading to significant out-of-plane motions.^{12,27} The latter increase third coordination sphere distances of carbons in cis-configuration (see above) and decrease third coordination sphere distances of carbons in trans-configuration. The corresponding signatures in a Δ PDF are negative peaks at the positions of the two cis and trans maxima (~ 3 Å and ~ 4 Å, respectively) of the third coordination sphere (see Fig. 2a) and a positive peak in the gap between the maxima of the third coordination

sphere (3.4 Å), overlapping with the observed γ signature. However, the early signature in the γ region as observed in the present study agrees only partially with this expectation: We observe a clear positive signature at 3.4 Å and a weak negative signature at 4 Å, which is close to the noise level in the experimental data, but clearly visible in the simulations (see **Supplemental Figure S1**). However, but a corresponding negative signature at smaller distances, around 3 Å, is missing. Thus, the early onset of the γ signature must exclusively originate from a distance reduction of larger third coordination sphere distances in trans configuration (see above) and distances from higher coordination spheres. An exclusive reduction of third coordination sphere distances in trans-configuration can only be consistent with out-of-plane motion of the (C₁₀) “reporter” carbon of the methyl substituent (e.g., (C₃,C₁₀)). The observed effect, a distance decrease from 4 Å to 3.4 Å, cannot be caused by a shrinking of the (C₅-C₁₀) bond distance. Such a motion would significantly shorten the bond distance far into the repulsive part of its potential. Additionally, it would generate a corresponding signature with similarly early onset in the α -region of the Δ PDF. An in-plane shrinkage of the (C₃-C₁₀) distance over several bonds would again also cause decreases of third coordination sphere distances in cis-configurations. Out-of-plane motion of the isopropyl group would also reduce distances in cis-configuration (e.g., (C₈,C₆)) which is not supported by our data.

Our simulations give further evidence for such a motion as visualized at the example of the (C₃,C₁₀) distance in **Fig. 3c**. At 100 fs after photoexcitation, the methyl ring substituent shows significant out-of-plane displacement (see **Fig. 4c**, green for the out-of-plane angle and the structures in **Fig. 3c**), which leads to a reduction of the (C₃,C₁₀) distance from 3.9 Å to 3.255 Å while the (C₃-C₄) and the (C₃,C₅) distances do not yet show substantial displacements. The (C₃-C₄) distance only shows considerable enlargement after 150 fs. With the (C₃-C₄) distance increase, the (C₃,C₁₀) pair contributes to the Δ PDF at higher distances. The contributions to the γ regime are taken over by the (C₃-C₄), (C₃,C₅), and other carbon pairs not highlighted in **Fig. 3c**.

- *Moreover, it seems to me that the out of-plane of the C2 is discarded making the hypothesis that it moves as a rigid body. In principle, it is possible that opposite movements of C8 and C7 might compensate the supposed shortening of the distances. Is it true? Can they be ruled out? Are they ruled out on the grounds of the results of the simulation?*

We believe that such a motion can be ruled out on the grounds that extreme bond distances and angles would have to be assumed for the bonds of C8 and C7 with C2 to achieve the experimentally observed signal, i.e., an out-of-plane bending of C2 alone by almost 60 degrees.

- *Finally, I have to highlight that along all the text the authors very often use atom labels, and the only figure that assisted me to understand this discussion and many other ones, is the small chemical structure with labels reported as an inset of Figure 2.*

We apologize for this inconvenience. However, we would like to point out that Figure 1 also contains atom labels. In the revised version of Fig. 2 we increased the size of the inset as much as possible. Additionally, we added atom labels to Fig. 4 (see below).

- 4) *It took me some time to understand also the second part of page 9, starting with “Our simulations..”. The authors mention a decrease of the distance from 3.9 to 3.2 Ang. After a while I understood this should refer to the very small deviation toward the left of the blue line in Figure 3c at times from 0.05 to 0.1 ps. As far as I can see from the figure, that line does not seem to go below 3.5 Ang, and this was confusing me. More in general, since practically this is the only detail of panel 3c discussed in the text, I’m wondering if the authors could not choose a different scale better helping the readers to follow their discussion.*

The referee refers to an error in the manuscript, for which we apologize. The decrease does not take place from 3.9 to 3.2 Angstrom, but from 3.9 to 3.55 Angstrom, in agreement with Figure 3c. To correct the mistake, we change the following sentence in the second paragraph of page 8:

At 100 fs after photoexcitation, the methyl ring substituent shows significant out-of-plane displacement (see Fig. 4c, green for the out-of-plane angle and structures in Fig. 3c), which leads to a reduction of the (C₃,C₁₀) distance from 3.9 Å to 3.255 Å while the (C₃-C₄) and the (C₃,C₅) distances do not yet show substantial displacements.

We address the referee’s comment on panel c in Figure 3 by increasing its size and optimize the horizontal axis limits (see above).

- 5) *Figure 4 left panel: Once printed the blue lines are hardly distinguished from the black (gray?) lines. It is maybe a problem of my printer, but I recommend the authors to better check the readability of the figure in printed form.*

We thank the reviewer for their suggestion and lightened the blue for the S1 trajectories to make them more distinguishable (see below).

- 6) *Do I understand properly that the authors have no experimental clue on the timescale of the internal conversion? The information that it takes at least 100 fs only comes from simulations, am I right? How confident can we be of this prediction? Some additional discussion is maybe worthy.*

The referee is correct in the sense that our observable is not sensitive to the electronic structure change which is induced in the molecule through internal conversion. However, it is very well-established that alpha-terpinene and related systems like cyclohexadiene and alpha-phellandrene open the ring quasi-instantaneously upon internal conversion to the ground state (see refs. 7, 8, 12, 13, 15, 17-20, 22 of the manuscript). Therefore, there is strong indirect evidence in the experimental data alone for this ring-opening timescale through the pump-probe delays of the alpha and beta features in the difference pair distribution functions. The 100-150 fs internal conversion

timescale is in overall agreement with time-resolved spectroscopic studies, which provide more direct evidence for the internal conversion timescale (see refs. 7 and 20). To clarify this point, we add the following sentence to the last paragraph of page 6: **The observed excited state lifetime is in good agreement with previous spectroscopic studies.**^{7,20}

- *7) In Figure 4C, the definition of the planarization and deplanarization coordinates in the insets of Figure 4 b and c is a bit hard to follow. This is partly due to the fact that the atom labels are not reported and should be consulted in Figure 2, the vectors are small and their direction not perfectly appreciable on that small scale. Moreover, the variable colours of the symbols of the sums defining phi and psi are maybe chosen to suggest something that may be not captured by the reader. I do understand that behind these figures there is a lot of work and that is hard to do better. However, at least I suggest the readers to exploit also SI, taking more space and providing larger (and maybe different) figures to help the reader to understand these definitions*

We agree with the referee that the definition of the planarization and deplanarization coordinates can be improved. The color-coding of the sums defining psi and phi refer to the vector arrows in the insets. We believe that this makes sense and is hopefully now better comprehensible with enlarged insets. Therefore, we changed figure 4 in the following aspects(revised figure shown below):

- We added number labels to the geometry in Fig. 4a. and mention it in the figure caption with the following sentence:

The atom labeling is shown in the inset.

- We increased the size of the geometry insets defining the planarization and deplanarization coordinates in Fig. 4 b and c

- We changed the appearance of the vector arrows in the insets to make their directions better visible.

- 8) At the beginning of page 13, I suggest the authors to be more explicit on what they mean by “correlation” and “anti-correlation” between the two considered structural parameters.

We appreciate referee 3’s suggestion and modify the text accordingly (page 10, last paragraph):

Corresponding projections onto the (C_3-C_{10}) distance and ψ are plotted in the lower row of Fig. 5. Specifically at the onset of population transfer to the ground state (100 fs delay), the projection shows significant deformation from an initially round shape into the diagonal direction due to an anti-correlation between the (C_3-C_{10}) distance and ψ (lower row), i.e., a (C_3-C_{10}) distance decrease correlated with a ψ increase specifically at the onset of population transfer to the ground state (100 fs delay). In contrast, there is neither strong correlation (distance increase with angle increase) nor anti-correlation for the (C_3-C_4) distance and ψ (upper row).

- 9) As far as I understand, in Figure 5, the grey spots indicate points at which the nonadiabatic transition takes for different trajectories (Gaussian WP). I don’t understand how it is possible that these points are all outside the region visited by the wavepacket (the red contour lines).

Referee 3 understands our figure correctly. The grey points visualize the coordinates in our projection where we observe population transfer to the ground state. The main reason why not all points overlap with the contour plots of the wavepacket in Fig. 5 is

that we only show 4 snapshots over a 150 fs range. The referee's question convinced us that it would be instructive to show the whole dynamics in the projection of Fig. 5 as a supplementary movie. The movie shows a much stronger overlap between the contour plot from the simulated wavepacket and the grey points. Some of the points still do not seem to be visited by the wavepacket. This due to the interaction of the weak edges of the Gaussian wavepacket with regions of strong nonadiabatic coupling. Our contour plot is a compromise between representations of strong and weak parts of the wavepacket. The contour lines are not representing these weak parts of the wavepacket very well. We refer to the new movie in the caption of Fig. 5:

For an animated version of Fig. 5 see **Supplementary Movie 1**.

Additionally, we mention it in the following sentence in the last paragraph of page 10: The upper row of Fig. 5 shows two-dimensional projections of the simulated excited state wavepacket density (red contour lines) onto the (C₃-C₄) distance and the deplanarization angle ψ from Fig. 4c at different delay times (see **Supplementary Movie 1** for an animated version of Fig. 5).

- 10) *The conclusion reported in the last three sentences of the manuscript is really interesting. Am I right if I state that, at the end, this conclusion is reached only thanks to the nonadiabatic simulations? I mean simulations are fully consistent with the available experimental observables, but the experimental observables by themselves have not a unique interpretation leading to the reported conclusion. Right? This could be made a bit more evident, and maybe one-two lines could be added to discuss if this leaves some uncertainty in the conclusions (e.g. if possible alternative interpretation exist).*

We only partially agree with referee 3 in this point. The experiment on its own gives clear evidence for a separation of timescales between the methyl group out-of-plane motion and the bond dissociation. We agree with referee 3 that the interpretation of the out-of-plane motion as a signature of rehybridization and bond alternation dynamics in the presence of the still unbroken sigma bond would stand on much shakier grounds without the theory support. On the other hand, the same conclusions could have been reached based on the theory alone.

There are, however, two important aspects which point out the need for the combination of experiment and theory:

-The quantitative agreement between experiment and theory is a strong confirmation of the validity of the theory results.

-Based on our observations in the present manuscript, we went back to the simulations of the prototypical 1,3-cyclohexadiene system, which we had performed for Ref. 15. These simulations show essentially the same effect, albeit on a faster timescale. The effect is likely included in many if not all of the previously published simulations on cyclohexadiene and related compounds. These simulations exhibit a high-dimensional information content requiring projections on appropriate lower-dimensional observables to extract mechanistic information. Experimental observables can serve exceptionally well for such projections and, thus, can be key in gaining a more detailed

understanding of the reaction dynamics.

We believe that this last point is important and reflected it in the sentences we added to the conclusion paragraph based on referee 2's comments:

The presence of the methyl "reporter" group of α TP in our chosen experimental observable, ultrafast electron diffraction, merely enables its investigation.

In summary this is a beautiful work and I strongly support its publication in Nature Comm. The authors made a lot of work to compact the text and increase as much as possible the information carried by the beautiful figures. I think in some cases they were too ambitious, and the figures are a bit hard to digest. Along the same lines, maybe, compatibly with the space, some additional, a bit more didactic, discussion might make the paper more accessible to a wide readership. These changes can be done in a minor revisions.

We thank referee 3 for their support and believe that the above outlined revisions completely address the referee's concerns.

A typo: Page 6 "largerdistances"

We corrected the typo.

REVIEWERS' COMMENTS

Reviewer #2 (Remarks to the Author):

The authors have addressed my concerns, I recommend the paper be accepted as is.

Reviewer #3 (Remarks to the Author):

I read the revised version. I think the authors further improved the quality of their manuscript and I confirm that I strongly support its publication in Nature Communications.

Response to referee comments: We format the referee comments in normal text and our response in *italic*.

Reviewer #2 (Remarks to the Author):

The authors have addressed my concerns, I recommend the paper be accepted as is.

We thank reviewer #2 for their support.

Reviewer #3 (Remarks to the Author):

I read the revised version. I think the authors further improved the quality of their manuscript and I confirm that I strongly support its publication in Nature Communications

We thank reviewer #3 for their support.